# Disposable ultrasound-sensing chronic cranial window by soft nanoimprinting lithography

Hao Li[1,5], Biqin Dong[1,2,5], Xian Zhang[1,3,5], Xiao Shu[1], Xiangfan Chen[2], Rihan Hai [2], David A. Czaplewski [4], Hao F. Zhang [1] & Cheng Sun [2]

Chronic cranial window (CCW) is an essential tool in enabling longitudinal imaging and manipulation of various brain activities in live animals. However, an active CCW capable of sensing the concealed in vivo environment while simultaneously providing longitudinal optical access to the brain is not currently available. Here we report a disposable ultrasound-sensing CCW (usCCW) featuring an integrated transparent nanophotonic ultrasonic detector fabricated using soft nanoimprint lithography process. We optimize the sensor design and the associated fabrication process to significantly improve detection sensitivity and reliability, which are critical for the intend longitudinal in vivo investigations. Surgically implanting the usCCW on the skull creates a self-contained environment, maintaining optical access while eliminating the need for external ultrasound coupling medium for photoacoustic imaging. Using this usCCW, we demonstrate photoacoustic microscopy of cortical vascular network in live mice over 28 days. This work establishes the foundation for integrating photoacoustic imaging with modern brain research.

---

[1] Biomedical Engineering Department, Northwestern University, Evanston, IL 60208, USA. [2] Mechanical Engineering Department, Northwestern University, Evanston, IL 60208, USA. [3] Department of Ophthalmology, Tongji Hospital, Tongji Medical College, HuaZhong University of Science and Technology, 430030 Wuhan, Hubei, China. [4] Center for Nanoscale Materials, Argonne National Laboratory, Argonne, IL 60439, USA. [5] These authors contributed equally: Hao Li, Biqin Dong, Xian Zhang. Correspondence and requests for materials should be addressed to C.S. (email: c-sun@northwestern.edu)

Chronic cranial window (CCW) has been widely used to provide optical access to the brain cortex for longitudinal imaging over an extended period of time, while preserving the physiological environments of the brain[1,2]. The simplicity and reliability of the technique make it a popular choice for studying small animals. A single surgical procedure with fast recovery of the animal allows in vivo optical imaging on the same subject over an extended period of time. A wide range of optical imaging modalities, such as single-photon and multi-photon fluorescence microscopy[3], optical coherence tomography[4], photoacoustic microscopy (PAM)[5], and laser speckle contrast imaging[6] have been used with CCW for longitudinal brain imaging[7]. Recently, researchers also incorporated CCW with optogenetic tools to achieve selective optical stimulation and suppression of neuron activities in live animals[8].

Despite these successes, the surgical implantation of CCW precludes direct physical access to the brain other than optical access. Thus, studies that require chronic imaging and direct cellular recording or manipulation have to rely on sophisticated surgeries and window designs[9–11]. Transforming CCW into an active sensing device would provide a highly desirable solution for conducting additional long-term studies, while still preventing contaminating the brain and keeping the animal in a healthy condition. However, such a device is not currently available.

Here, we report an active CCW with integrated ultrasound sensor. As shown in Fig. 1, we develop a low-cost soft nano-imprint lithography (sNIL) method to fabricate disposable ultrasound-sensing CCW (usCCW) that compromises a transparent microring resonator (MRR)-based ultrasonic detector being integrated to the inner surface of the CCW. Fabricating usCCWs using sNIL is accomplished via a simple stamping step using PDMS (Polydimethylsiloxane) soft mold, eliminating the need for sophisticated nano-fabrication facilities. Using the sNIL process, we demonstrate more than 10-fold improvement in the sensitivity of the MRR ultrasonic detector. Encapsulating MRR inside an acoustic impedance matched protection layer further improves its reliability for in vivo applications over the observed period of 28 days. Moreover, the use of the sNIL significantly reduces the fabrication cost to be less than a dollar per device, which makes the usCCW suitable for one-time use upon surgical implantation. The functions of the active usCCW is validated experimentally through longitudinal photoacoustic microscopy (PAM) imaging of cortical vasculatures in live mice over 28 days.

## Results

**Fabrication of MRR detectors by soft-nanoimprinting.** We previously developed an optically transparent MRR ultrasonic detector with demonstrated detection bandwidth of over 250 MHz and a noise equivalent pressure of 6.8 Pa[12,13]. Although their distinct optical transparency and miniaturized form-factor offer unique advantage for potential integration with the CCW, they are not readily for the in vivo studies yet due to the following reasons. Firstly, the device is designed to operate in purified water[12,13]. When it is exposed to in vivo physiological environment, we discovered that its optical resonance diminished rather rapidly due to contaminations from, for example, cerebrospinal fluid and blood. Secondly, the sensitivity of the MRR detector needs to be further improved to accommodate the potential signal decay during the long-term studies. Finally, the MRR detectors are fabricated using the time-consuming electron beam lithography process with very low yield. It becomes inhibitive expensive to integrate MRR detector to the CCW as it is not possible to reuse the MRR detector once it has been integrated to the CCW.

In addressing the aforementioned issues, we optimize the usCCW to (1) improve the sensitivity by developing a sNIL process and (2) encapsulate the MRR to isolate it from potential contaminants. The fabrication flow is illustrated in Figs. 2a-k. We first spin-coat 200-nm photo-resist on the silicon substrate with an 80 nm-thick oxide layer. After patterning using the electron beam lithography process, the developed pattern in the photo-resist is transferred into the $SiO_2$ layer using reactive ion etching (RIE) process. The pattern is further transferred into silicon using deep reactive ion etching (DRIE) process to fabricate the silicon master mold containing high-aspect-ratio features. After

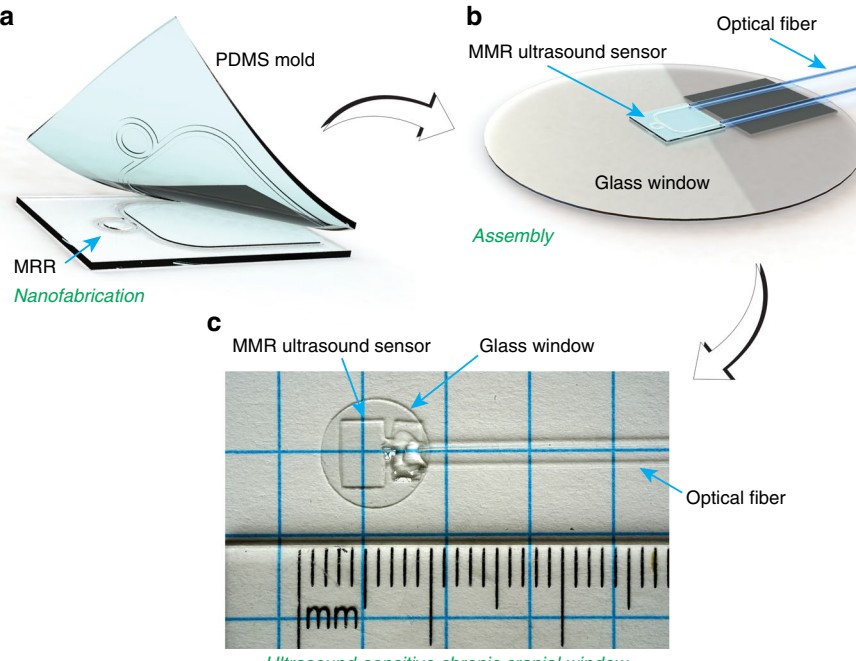

**Fig. 1** Process to fabricate usCCW: (**a**) Nanofabrication of MRR on transparent substrate using sNIL process; (**b**) assembling an MRR-based ultrasonic detector with matching optical fibers, onto a circular cover slide (8 mm in diameter); and (**c**) finally assembled usCCW device

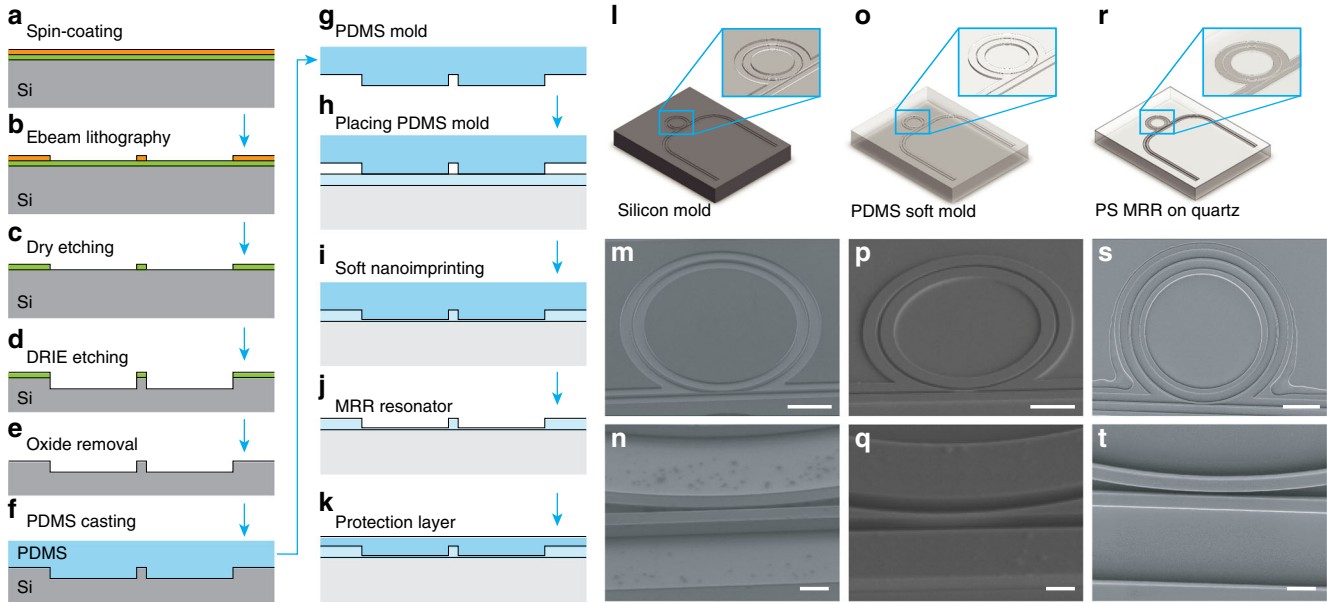

**Fig. 2** Fabrication process of the MRR ultrasonic detectors. The fabrication procedure consists of: (**a**) spin-coating of 200 nm photoresist; (**b**) patterning electron beam lithography process; (**c**) RIE pattern transfer from photo-resist into $SiO_2$ layer; (**d**) DRIE pattern transfer from $SiO_2$ layer into silicon substrate with high-aspect-ratio features; (**e**) removal of the $SiO_2$ layer; (**f**) casting the PDMS soft mold via thermal curing of the PDMS precursor; (**g**) peeling off PDMS soft mold from silicon master mold; (**h**) placing PDMS soft mold over a 400-nm-thick PS thin film spin-coated on a quartz coverslip; (**i**) molding process by heating the polystyrene film over its glass transition temperature; (**j**) after cooling down to room temperature, the PDMS stamp is peeled off; finally, (**k**) a 5-μm UV-curable PDMS thin film is spin-coated on the MRR waveguide and then cured by excessive UV exposure. The illustration of (**l**) silicon mold, (**o**) PDMS soft mold, and (**r**) fabricated MRR on quartz substrate and the magnified views shown as inset. The scanning electron microscope images of (**m**) silicon mold, (**p**) PDMS soft mold, and (**s**) MRR device. Their corresponding magnified image are shown as (**n**), (**q**), and (**t**), respectively. Scale bars: (**m**, **p**, **s**) 20 μm, (**n**, **q**, **t**) 2 μm

removing the residual $SiO_2$ layer, we perform thermal oxidation and wet etching processes to further reduce the surface roughness of the silicon master mold. The fabricated silicon mold is schematically illustrated in Fig. 2l and the corresponding scanning electron microscope image of the fabricated silicon master mold and the magnified view are shown in Fig. 2m, n, respectively.

The fabricated silicon mold is used to cast the PDMS soft mold by coating a thick layer of PDMS precursor on the surface of the silicon master mold. Upon thermal curing, the solidified PDMS soft mold is peeled off from the silicon mold. The fabricated PDMS soft mold is illustrated in Fig. 2o and the corresponding SEM images are shown in Figs. 2p and q. The fine details of the waveguides in the silicon mold are faithfully reproduced into the PDMS soft mold. One silicon master mold can be used repeatedly to fabricate many replicas of reusable PDMS soft molds to minimize the manufacturing cost.

After successful fabrication of the PDMS soft mold, we use it to fabricate the MRR device in the sNIL process. A 400 nm-thick polystyrene (PS) layer is spin-coated on a quartz substrate. The PDMS soft mold is placed onto the PS film. The PS film is heated over its glass-transition temperature to 125 °C for 60 min on a hot plate. The mold filling by molten polystyrene is driven by the capillary force. The mold design is optimized to effectively displace the molten PS during the molding process to minimize the thickness of the residual layer, while retaining the optimal mold filling (see Supplementary Note 1). After cooling down to room temperature, the PDMS stamp is peeled off, resulting a well-defined MRR device on the quartz substrate, as illustrated in Fig. 2r. The SEM image of the fabricated MRR device is shown in Fig. 2s and the magnified view shown in Fig. 2t. Finally, a 5 μm-thick UV-curable PDMS thin film is spin-coated on the MRR waveguide and then UV-cured to serve as a protection layer (See Methods section for more details of the fabrication process).

Although nanoimprinting process using rigid quartz mold has been previously reported to fabricate the MRR device on the silicon substrate[14], the sNIL process using soft PDMS mold offers multiple advantages. sNIL eliminates the need for high-pressure molding process and thus, the fabrication of usCCW can be conveniently accomplished in fumehood with a regular hot plate. It no longer requires a sophisticated nano-imprinting machine. The excellent elastic properties of the PDMS mold further mitigates the difficulty in demolding the high-aspect-ratio features. We also found the soft mold tends to conform to asperities in the surface and thus, increases the robustness against surface imperfections in comparing with the nanoimprinting process using rigid quartz mold. Potentially, it can be compatible with the curved substrate. In comparing with the previously reported soft lithography method also using PDMS soft mold[15], sNIL further eliminates the use of UV curing process so the low loss PS can be used to fabricate the usCCW with minimal absorption loss[14]. Collectively, they positively contribute to the increased fabrication yield and reproducibility. The process can be easily implemented by researchers with minimal capital investment and manufacturing cost.

As shown in Fig. 3a, the fabricated MRR consists of a micro-ring waveguide and a matching straight bus waveguide. The MRR supports optical resonance (whispering gallery modes) with a reasonably high-quality factor (Q-factor) that amplifies the ultrasound-induced deformation of the waveguide into a frequency-shift of the resonance modes in optical spectrum. The ultrasonic detection sensitivity is proportional to the Q-factor[12], which is inversely proportional to the energy dissipation through the waveguide internal loss and the coupling loss

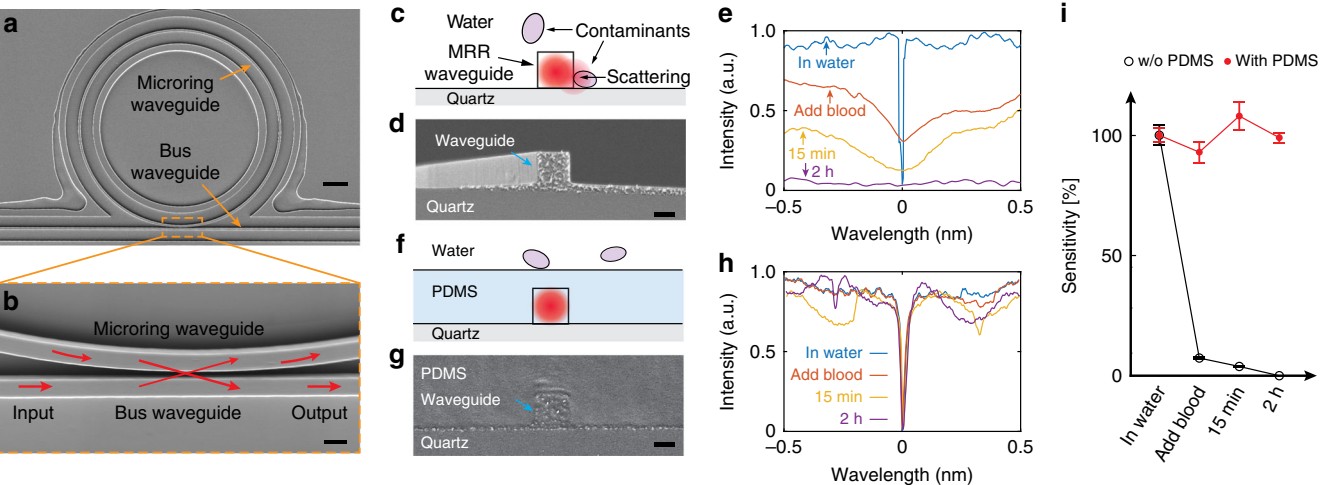

**Fig. 3** Protecting MRR ultrasonic detector for long-term in vivo imaging. **a** Scanning electron microscope (SEM) image of a MRR ultrasonic detector fabricated by sNIL. Scale bar: 10 μm; (**b**) Magnified view of the waveguide coupling region highlighted by the orange dashed box in **a**. Scale bar: 500 nm. **c** Schematic and (**d**) SEM image of a cross-section of the fabricated MRR. The schematic illustrates potential contaminants attached to the waveguide. Scale bar: 500 nm. **e** Optical resonance diminished in 2 h due to contamination when unprotected MRR is exposed to whole blood. **f** Schematic and (**g**) SEM image of a cross-section of the MRR protected by an additional PDMS layer. Scale bar: 500 nm. The schematic illustrates that contamination to the waveguide is prevented. **h** Optical resonance remained unaffected after the protected MRR is exposed to whole blood for 2 h. **i** Comparison of Q-factors of MRRs with and without the protection layer in whole blood. Error bars show the standard deviation of Q-factor when fitting with the experimentally measured resonance spectrum

between the micro-ring waveguide and bus waveguide[16]. Therefore, we develop a systematic strategy to reduce the energy dissipation in the MRR to maximize the ultrasonic detection sensitivity. The internal loss is determined by the absorption and scattering of the light propagating in the waveguide. We choose low-loss PS (M.W. 800–5000, Polysciences Inc.) to replace the previously used SU-8 (SU8-2005, MicroChem) as the base material to fabricate the waveguide to minimize absorption loss[14]. We also add the thermal oxidation and etching steps further reduces the surface roughness of the silicon master mold and thus, reduces the scattering loss. Furthermore, maximizing the Q-factor also requires the MRR to reach the critical coupling condition, where the coupling loss is fine-tuned to match the internal loss. However, in our previous study, it is difficult to fabricate the coupling region feature high-aspect-ratio gap separating the micro-ring waveguide and the bus waveguide using the direct electron-beam writing process. Often, the residuals found inside the gap compromises our ability to fine tune the coupling between the micro-ring waveguide and the bus waveguide, resulting the fabricated MRR detector with sub-optimal sensitivity and poor repeatability. To address this issue, we only use the electron beam lithography process to pattern the thin layer of photoresist and the high-aspect-ratio feature is subsequently transferred into the silicon substrate with high fidelity. We demonstrate the precise control of the gap width down to 100 nm with a high aspect-ratio of 8:1 (Fig. 2j). Such fabrication accuracy allows us to reliably reach the critical coupling condition to maximize the ultrasonic detection sensitivity. With these optimization steps, we achieve a Q-factor of $1.4 \times 10^5$, an order of magnitude improved from our previously reported MRR detectors fabricated by SU-8 photoresist using direct electron-beam writing process[12], resulting an estimated noise equivalent pressure of 0.49 Pa (see details in Supplementary Note 4).

**Introducing protection layer for long-term in vivo imaging.** We examine the stability of the MRR device after being exposed to physiological contaminants. The primary concern is that the high Q-factor optical resonance can be extremely vulnerable to

additional scattering losses due to surface contamination from biological tissues (Fig. 3c-e)[16]. The optical coupling between the micro-ring waveguide and the bus waveguide could also be compromised by contaminants filling the gap. We test the stability of unprotected MRR device by exposing it to bovine whole blood, which mimics hemorrhage frequently observed during in vivo experiments[1]. As shown in Fig. 3e, the Q-factor decreases rapidly immediately after the MRR is immersed in blood, and the optical resonance completely vanishes in 2 h. This is caused by the elevated scattering loss due to the deposition of contaminants from the blood to the waveguide surface. To overcome this problem, we encapsulate the MRR using a 5-μm-thick PDMS protection layer (Figs. 3f-h and Supplementary Fig. 1). The selection of PDMS as the protection layer is motivated by multiple reasons. First, it is a well-accepted biocompatible material[17]. Second, its refractive index of 1.40 matches closely with the refractive index of quartz substrate, which allows symmetric light confinement inside the waveguide to further reduce the propagating loss (see Supplementary Note 2 and Supplementary Fig. 2 for numerical simulation of optical waveguide modes). Third, the acoustic impedance of PDMS matches with water to minimize the ultrasound reflection loss[18] (see Supplementary Note 3 for calculation of acoustic reflection on PDMS-water interface. The 5-μm-thick PDMS cladding layer has negligible attenuation less than 0.6 dB of the ultrasound signal at 100 MHz[19]). Experimentally measured detection sensitivity and frequency response of MRR shows comparable responses with and without the PDMS protection layer (Supplementary Fig. 4). In contrast to the MRR without PDMS protection, the encapsulated MRR shows no obvious change in Q-factor over 2 h when being exposed to the whole blood (Fig. 3h). The quantitative comparison of the ultrasonic detection sensitivities shown in Fig. 3i further suggests that the MRR with PDMS protection layer can operate reliably in biological environments.

**Surgical implantation of usCCW and PAM cortical imaging.** To demonstrate that active CCW allows additional measurement capability other than optical access, we show that usCCW enables longitudinal PAM cortical imaging (Fig. 4a) for nearly one

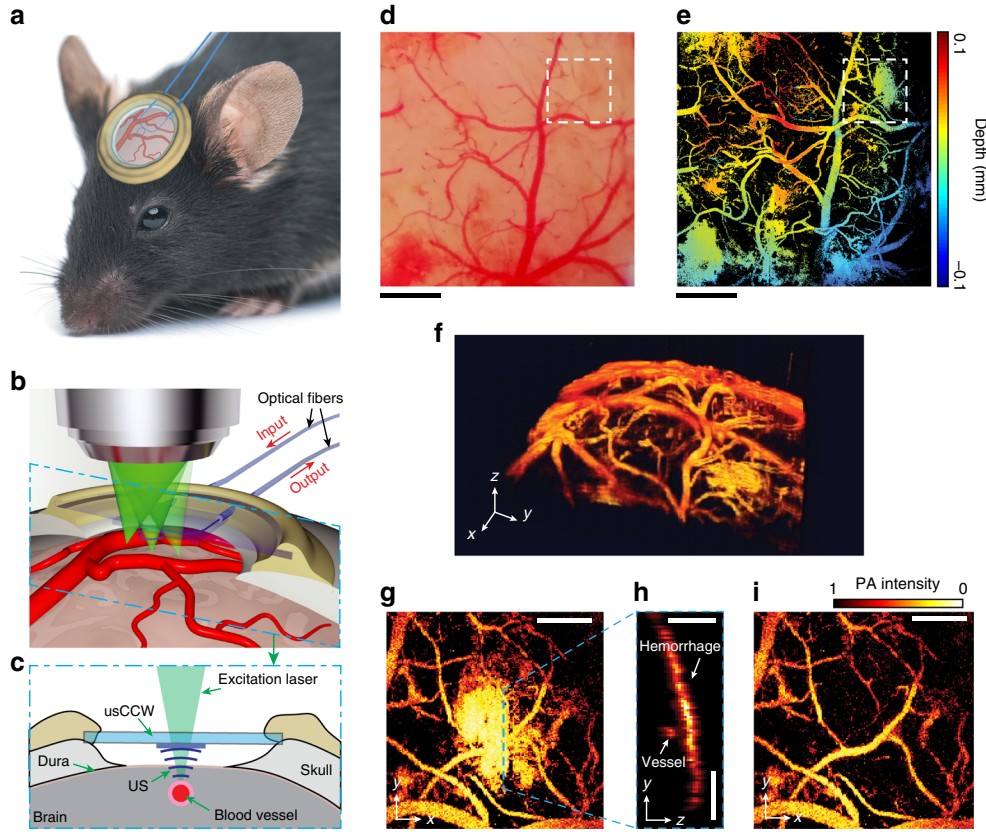

**Fig. 4** In vivo PAM cortical imaging using a usCCW. **a** The usCCW is surgically implanted on the mouse skull after craniotomy. The inset shows the physical dimension of the usCCW with MRR and fibers attached, which is optically transparent, with a total thickness of 250 μm and a total weight of less than 1 g. The MRR ultrasonic detector is attached on an 8-mm diameter circular substrate and the sensing light is coupled through a pair of 30-cm flexible optical fiber. **b** Illustration of optical scanning through the usCCW. To excite the MRR resonance, a narrow-band continuous-wave tunable laser (New Focus, TLB-6712, wavelength from 765 nm to 781 nm) is coupled into the bus waveguide after passing through a fiber polarization controller, and collected by a multimode fiber on the other end of the bus waveguide. **c** Optical excitation and ultrasonic detection geometry along the cross section highlighted in **b**. The space between the MRR and the dura is 1 mm and is filled with 0.5% agarose gel. We seal the usCCW with dental cement to prevent infection and leakage. **d** Brightfield optical microscopy image of the cortical region through the MRR. **e** Depth-encoded maximum-intensity-projection (MIP) PAM image of the same area. The whole image is stitched from 9 acquisitions due to the limited laser-scanning field of view[27]. **f** Three dimensional visualization of the vessel orientations and cortical curvacure. **g** PAM image of the hemorrhage area highlighted by the dashed box in **d** and **e**. **h** PAM B-scan image from the position highlight by the green dashed line in **g**, showing the hidden vessels beneath the hemorrhage area. **i** Visualizing vessels beneth the hemorrhage layer. Scale bars, (**a–b**) 0.5 mm and (**g–i**) 200 μm

month. Among existing optical microscopy modalities, PAM offers unique capability in microvascular imaging based on endogenous optical absorption contrast from hemoglobin[20,21]. PAM employs a short-pulse laser beam focused onto biological tissues. The absorbed laser energy leads to a transient thermo-elastic expansion and, subsequently, launches ultrasonic wave with a wide-range of frequency components, which can be detected to reconstruct optical energy deposition-based images.

Several groups have demonstrated the potential advantages of applying PAM to brain investigations[22,23]. The intact mouse skull has long been a major technical hurdle for high-resolution photoacoustic brain imaging. The underlying scattering of the incident laser beam and attenuation of acoustic signal due to acoustic impedance mismatch, respectively, hampers both lateral and axial resolution of photoacoustic imaging. Specifically, the strong scattering of the incident laser beam unfavorably broadens the focal spot size and thus reduce the lateral resolution. On the other hand, the large acoustic impedance mismatch at the skull-brain interface leads to strong attenuation of the generated photo-acoustic signals, especially the high-frequency signals above 30 MHz[24,25]. As a result, the best reported PAM imaging results

has a highest axial resolution of 30 μm using a 50 MHz ultrasound transducer[22,23]. Such a low axial resolution poses fundamental challenges in differentiating capillary networks in the skull, the durum, and the cortex. We further perform comparative studies of PAM brain image with and without the skull to better illustrate the remaining challenges in high-resolution photoacoustic brain imaging (details see Supplementary Note 5).

More recently, using cranial window in PAM further improves the spatial resolution by eliminating the optical scattering by the skull[5]. Despite the demonstrated potentials, PAM cortical imaging through cranial window has only been demonstrated in short-term studies, mainly constrained by piezoelectric transducers being used to detect ultrasound signals. Due to the size constrain and opaque nature, it is difficult to accommodate the piezoelectric transducer inside a cranial window[26]. Thus, the whole experimental setup needs to be immersed in a water tank to couple the ultrasound signal, through the cranial window, to the piezoelectric transducer[5]. The inherent complexity in the experimental procedure unfavorably compromise its success in long-term studies. Our usCCW device would provide an ideal solution to this problem.

As illustrated in Fig. 4a, we perform craniotomy and surgically implant the usCCW onto the skull with the MRR facing cortex for ultrasonic detection. The miniaturized MRR fitted well inside the narrow space between the dura and glass window (Fig. 4b). When conducting PAM imaging, the excitation laser beam illuminates through the usCCW (Fig. 4c) and scans across the region of interest on the cortex. The generated ultrasonic wave introduces a conformational alteration of the MRR, which is detected optically and correlated with the ultrasonic pressure[27]. The MRR is placed 1 mm away from the cortex to ensure a $360 \times 360\,\mu m^2$ PAM field of view at 166 MHz ultrasonic detection bandwidth according to the theoretical model we reported previously[25]. The extended field of view of $600 \times 600\,\mu m^2$ can be obtained at 100 MHz bandwidth. We fill the gap between the dura and the usCCW with agarose gel[28] for ultrasound coupling, which eliminates the need for a water tank and permitted long-term PAM imaging of live rodents.

We first test the MRR by imaging exposed cortical vasculature in mouse brain in vivo with the skull removed (see Methods for surgical procedure). We place the MRR detector between the objective lens of a commercial microscope (Olympus BX61) and the mouse brain (see Methods for detailed protocol and Supplementary Fig. 5 for the imaging setup). Figure 4d shows the bright-field optical microscopy image of the exposed cortex under the MRR, which verifies the clear optical access through our MRR. The corresponding depth-coded enface PAM image of the same area is shown in Fig. 4e, where detailed vascular morphology is visualized as a result of the strong absorption contrast from hemoglobin[22]. The PAM image shows much finer details of the small capillaries that cannot be resolved from the optical image (Fig. 4d). Our MRR detector achieves an ultrasonic detection bandwidth up to 280 MHz and an axial resolution of $2.12\,\mu m$[29]. The corresponding axial resolution in this study is estimated to be 3.57 μm based on the detection bandwidth of 166 MHz. Combining broad-band time-resolved ultrasonic detection with optical raster scanning[13], we are able to distinguish single capillaries in 3D as shown in Fig. 4f (see its animation in Supplementary Movie 1).

In addition to imaging blood vessels, PAM is also well suited to visualize temporary hemorrhage[1,30], which can be induced during craniotomy as a thin layer of residual blood covering the cortex that cannot be completely removed after rinsing. Such a hemorrhage region is highlighted by white dashed boxes in Figs. 4d and e. Although this thin residual blood layer is barely recognizable in the optical image (Fig. 4d), it can be clearly observed from the PAM image (Fig. 4e). The magnified PAM image of the hemorrhage region is shown in Fig. 4g. Although the blood overlaps with the vasculature in the maximum-intensity-projection PAM image, they can be well separated in depth resolved B-scan images (Fig. 4h) thanks to the high axial resolution offered by the MRR. As a result, we can further segment and remove the hemorrhage layer in the PAM image to reveal the vasculature beneath (Fig. 4i).

**Longitudinal PAM cortical imaging using usCCW.** Finally, we conduct long-term intravital PAM imaging of live mouse brains using our usCCW (Figs. 5a and b). We implant usCCW on mice in both thinned-skull and open-skull configurations. While our usCCW is compatible with both configurations, we find that thinned-skull configuration results in weaker ultrasound signal (Supplementary Note 6 and Supplementary Fig. 7) due to higher acoustic attenuation caused by the residual bone structure. Moreover, the skull regrowth and window occlusion lead to further attenuations of both optical and ultrasound signals. We

had to re-thin the skull after 20 days to minimize attenuations for longitudinal investigation.

The usCCW in open skull configuration is implanted on mouse's forehead following craniotomy[1,30] (see surgical details in Methods). Only a pair of optical fibers is attached to the usCCW, which imposes minimal constraint to mouse's free-motion (Fig. 5a and Supplementary Movie 2). At desired time point, we restrain mice on a home-built animal holder for PAM imaging and release them to their respective cage afterwards. By integrating MRR detector to the inner surface of the cranial window, we eliminate the need of a water tank for ultrasound coupling, which significantly simplifies the experimental procedure and reduced potential risk of infection from water contact (Fig. 5b). The magnified optical image of the implanted usCCW is shown in Fig. 5c. Our usCCW provides unobstructed optical access over a 6-mm-diameter circular area of the cerebral cortex. In this study, we perform PAM imaging over 28 days. Figure 5d shows a marginal reduction of the Q-factor from $4.2 \times 10^4$ on Day 0 to $3.6 \times 10^4$ on Day 28, which confirms the stability of usCCW for in vivo applications. The projection PAM images in Fig. 5e show the evolution of the cortical vasculature in the same area on selected days. Figures 5f-h further show the, respectively, pairwised comparison of the cortical vasculature between Day 0 and Day 2, Day 0 and Day 14, and Day 14 and Day 28 to better illustrate the observed neovascularization process. Their corresponding locations in the originally acquired PAM images are marked by the yellow dashed-boxes in Fig. 5e. We record a moderate hemorrhage occurred on Day 3, followed by gradual reabsorption from Day 6 to Day 10. Since the hemorrhage occurred on Day 3, it is unlikely to be caused by direct tissue damage during the craniotomy, but rather secondary to possible superficial trauma due to animal activity or spontaneous in nature. The subsequent neovascularization that is witnessed during the days following hemorrhage clearance is consistent with the pathophysiology described in traumatic brain injury models[31,32] and known angiogenesis models following intracerebral hemorrhage clearance[33]. Injury to the cortex activates a cascade of molecular events that promotes the release of pro-angiogenic factors, such as vascular endothelial growth factor, which activate endothelial cells to initiate neovascularization post intracerebral hemorrhage[34]. It subsequently facilitates erythrocyte efferocytosis and allows for local tissue repair.

## Discussion

In contrast to the existing CCW being primarily designed as a passive optical window, we demonstrate that usCCW is a functioning imaging component by integrating a MRR ultrasonic detector. We achieve longitudinal PAM imaging of mouse cortex in vivo over 28 days. The usCCW provides unobstructed optical access to the cerebral cortical region while simultaneously enables 3D isometric PAM imaging at single-capillary resolution. As the results, we demonstrate a smart cranial window based photoacoustic microscopy with an axial resolution of ~4 μm, which is the highest axial resolution ever achieved in photoacoustic brain imaging in vivo. Because our window itself is an ultrasonic detector located in close proximity to the brain tissue, the high axial resolution is achieved without sacrificing the optical focusing or the photoacoustic signal strength. The demonstrated 8-fold improvement in axial resolution makes it possible to clearly distinguish capillaries in the durum and cortex and thus, offers promising opportunities for fundamental biological investigations.

The usCCW not only allows chronical PAM imaging; it is also compatible with other established optical microscopy modalities and potential optogenetic manipulation. For example, integrating PAM imaging of hemodynamics in the cortex with two-photon imaging of neuron activities could provide new insight of neurovascular coupling[35]. Long-term monitoring of cerebral blood circulation can

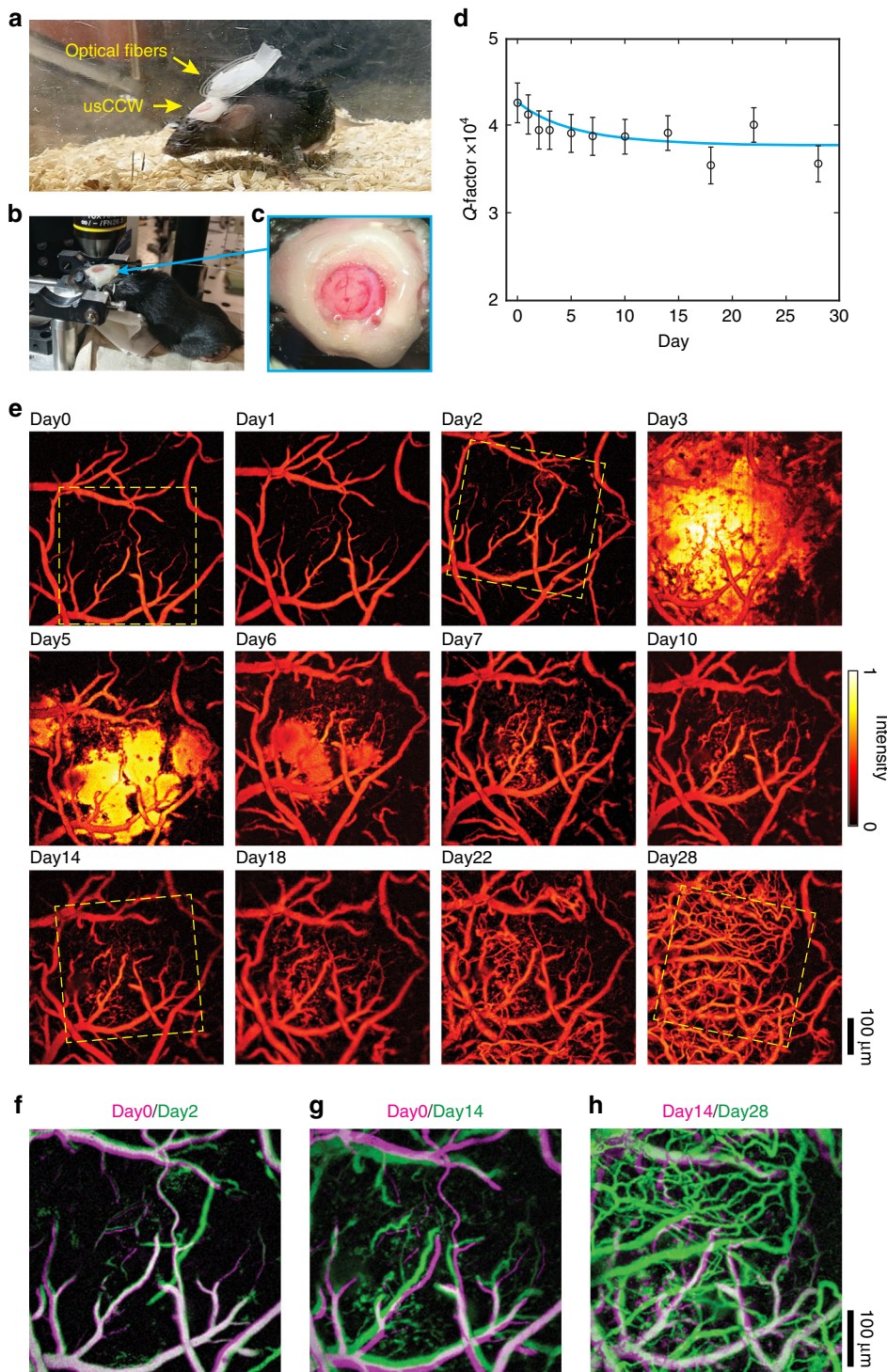

**Fig. 5** Longitudinal cortical PAM through the usCCW. **a** Photograph of a free-moving mouse in the breeding cage after the implantation. **b** Photograph of the mouse mounted under the microscope during the imaging. **c** Maginfied optical image shows the implanted transparent usCCW and the clear optical view of the cerebral cortex. **d** Q-factor shows marginal reduction from $4.2 \times 10^4$ to $3.6 \times 10^4$ over the 28-day-period. Error bars show the standard deviation of measured results. **e** Maximum amplitute projections of PAM images show cortical vasculature in the same area over a 28-day-period. Medium post-surgical bleeding occurs at Day 3 and the hemorrhage is gradually cleared in 4 days. Aggressive neovasularization is also clearly observed. The pairwised comparison of the measured cortical vasculature between (**f**) Day 0 and Day 2, (**g**) Day 0 and Day 14, and (**h**) Day 14 and Day 28. Superposition of two images rendered in distinct megenta and green colors is used to provide direct visualization of the neovascularization process. The overlapping regions are therefore rendered in white color

further the investigations on various cerebral disorders including as brain tumors[36], stroke[33,37], traumatic brain injury[32], and even neurodegenerative diseases[38]. Moreover, a light-weight wearable imager can potentially be developed using usCCW to image non-restricted, free-moving animals[39,40]. Although we only demonstrated adding ultrasound sensing capability to CCW in this work; sNIL process can be further applied to other optically transparent devices, such as microfluidic circuits or even the planar metasurface lenses, to be integrated with CCW for more comprehensive brain cortical investigations and manipulations.

## Methods

**Fabricating transparent MRR ultrasonic detectors using sNIL**. The fabrication processes comprise four steps: (1) fabrication of silicon master mold, (2) making PDMS soft stamps by pattern transfer, (3) soft-nanoimprinting of MRRs on quartz substrates, and (4) coating the PDMS protection layer.

To fabricate silicon master mold, we first deposited an 80-nm-thick $SiO_2$ layer (plasma enhanced chemical vapor deposition, Oxford Instruments PlasmaPro 100) on a silicon wafer (0.25 mm thickness, 100 orientation) as a silicon etching mask. A strike step lasting 5 seconds ignited the plasma using 20 sccm of $N_2O$, 8.5 sccm of $SiH_4$, at 5 mTorr and 100 °C, 50 W of RF power and 1200 W of inductively coupled plasma (ICP) power. The deposition is performed for 6.5 min after reducing the pressure to 2.5 mTorr and turning off the RF power. We then spin-coated the electron beam resist (Gluonlab gL 2000–12, diluted 1:1 with anisole by volume) on the wafer at 2000 rpm for 45 seconds and then baked at 180 °C for 3 min, yielding a thickness of 200 nm. The pattern is defined by exposing the wafer in a JEOL JBX-9300FS electron beam lithography (EBL) wafer system with an electron acceleration of 100 kV and a dose of 800 μC/cm$^2$. After EBL process, the wafer is developed in Xylene at −5 °C for 45 seconds and then rinsed with isopropyl alcohol and blown dry with $N_2$.

Following development, the resist pattern is transferred into the deposited $SiO_2$ layer by reactive ion etching (RIE, Oxford Instruments Plasmalab 100) for 5 min with 50 sccm of $CHF_3$, and 2 sccm of $O_2$ at 10 mTorr and 20 °C with 50 W of RF power. After the pattern transfer, the remaining resist is removed in a March V plasma system for 2 min using 24 sccm of $O_2$ at 160 mTorr and room temperature and 200 W of RF power. Then the pattern is transferred from the $SiO_2$ hard mask into the underlying silicon using the aforementioned Oxford RIE system. In order to remove any silicon dioxide that may have formed during resist removal in the pattern area, a 5-s etch is performed using 20 sccm of $Cl_2$, at 10 mTorr and 20 °C, with 300 W of RF power and 1000 W of ICP power. Then the main etching is performed in the same system for 7 min using 50 sccm of HBr and 2 sccm of $O_2$, ramping the pressure after the plasma ignites from 20 mTorr to 12 mTorr over 15 s, at 20 °C, with 100 W of RF power and 250 W of ICP power.

We further smoothed the mold surface by thermal oxidizing the silicon surface for 20-nm deep (Tystar furnace, 25 min dry oxidation at 1000 °C), and removing the $SiO_2$ layer by buffered oxide etchant[41]. The mold is then treated in TFOCS ((Tridecafluoro-1,1,2,2-tetrahydrooctyl)−1-trichlorosilane) vapor for 30 min to create a hydrophobic surface[41].

To fabricate PDMS soft stamps, we poured the PDMS precursor (Dow Corning Sylgard 184, 10:1 stir-mixed of two parts and degassed for 30 min) on the silicon master mold. After a 20-min consolidation at 100 °C, the cross-linked PDMS with transferred MRR patterns is carefully peeled off from the silicon mold.

To transfer the MRR pattern onto quartz substrates, we first spin-coated a 400-nm thick thin film of polystyrene on a quartz coverslip (University Wafer, 22 × 22 mm, 250-μm thickness) using 8% polystyrene/toluene solution at 2000 rpm. We then placed the PDMS soft stamp onto the polystyrene film. After squeezing out the air between stamp and coverslip, coverslip along with the stamp is placed on a hot plate and heat to 125 °C for 60 min. The melted polystyrene can fill MRR waveguide pattern on the PDMS soft mold by capillary force. We then let the hot plate cool down to room temperature and peeled off the PDMS stamp. Each PDMS stamp can be reused for 3 to 5 times.

To coat the PDMS protective layer, we increased the surface energy of the polystyrene MRR using $O_2$ plasma treatment (South Bay Technology PC 2000 Plasma Cleaner, 50 sccm, 50 mTorr, 50 W, 10 s). We then spin-coated a -5μm thin film of UV-curable PDMS (KER-4690, Shin-Etsu, 1:1 stir-mixed and degassed for 10 min) at 8000 rpm for 5 min. The film is then cured by excessive UV exposure (375 nm, 10 W, 30 min).

**Animal handling, craniotomy, and CCW implantation**. All experimental protocols in this study were approved by the Animal Care and Use Committee of Northwestern University. We anesthetized the mouse (C57BL/6, 20 to 25 g) using isoflurane/air mixture (4% V/V during induction and 1.5–1.8% during surgery) at a flow rate of 0.5 L/min. The mouse core temperature is kept at 37 °C using a homeothermic blanket system with feedback control from a temperature probe inserted into the mouse rectum (FHC, Stoelting Co.). Heart rate, respiration rate (if ventilated) are continuously monitored using manual record. In addition, we checked the paw pinch reflexes every 15 min. To assure proper hydration, warm lactated Ringer's saline will be administered subcutaneously to animals. The corneas are protected by artificial tears.

We administered subcutaneously dexamethasone to reduce cerebral edema, and then immobilized the anesthetized mouse on a customized stereotaxic frame. We shaved the left head and disinfected the surgical region by betadine and ethanol for 3 times. We incised the skin and exposed the skull region of interest. We drew a cycle of about 4 mm in diameter with the dental drill. For the thinned-skull surgery, we thinned the skull inside the circle until brain vasculature can be clearly seen. For the open-skull surgery, we further thinned the skull at the edge of the circle. We stopped drilling when a very thin layer of bone is left on the edge, and then lifted skull away with a thin-tip forcep.

The usCCW with about 30-cm coupling optical fibers is then placed on the top of the thin skull or the open skull area with the MRR face done. We placed a 0.8-mm-thick ring-shape plastic spacer beneath the usCCW to separate the MRR detector from brain to increase the field of view, and filled the gap with 0.5% agarose gel. Finally, we sealed the usCCW and the spacer by dental cement to prevent infection and leakage of water inside the cranial window. After the implantation of the usCCW, subcutaneous injection of Buprenorphine-SR-Lab is used to relieve postoperative pain.

**PAM imaging setup and procedures**. The PAM image on the cortex of an anesthetized mouse is taken under a commercial upright microscope (Olympus BX61). Photoacoustic signal is excited by a 532-nm Nd:YAG nanosecond pulsed laser (Elforlight SPOT-10–500–1064 with a BBO crystal for light frequency doubling, 10-nJ per pulse energy, 1-ns pulse duration). A dual-axis galvo-mirror assembly (Nutfield Technology) is used to raster scan the laser beam, and a matching 1:3 Keplerian telescope further coupled the laser beam to the back aperture of the objective lens (Olympus, 4×, NA 0.1) through the side port of the microscope body. The detection of the PA signal is accomplished by using the MRR ultrasonic detector integrated on the CCW. To excite the MRR resonance, a narrow band continuous wave (CW) tunable laser (New Focus, TLB-6712, wavelength 765 nm to 781 nm) is coupled into the bus waveguide after passing through a fiber polarization controller, and collected by a multimode fiber on the other end of the bus waveguide. The transmission intensity is recorded using a photodetector (Newport, 2107-FC). Once the resonance is found, the output wavelength of the narrow band tunable laser source is set at the wrist of the resonance dip. The output fiber is switched to an avalanche photodiode (APD, Hamamatsu c4777, bandwidth 10 kHz to 100 MHz). The recorded signal is then amplified (Mini-circuits ZFL500NL+, 500-MHz bandwidth), digitized, and recorded using a Digitizer (CobralMax, GaGe).

During imaging, we place the anesthetized mouse under the microscope and clean the usCCW. The input fiber is linked to the CW tunable laser using a mechanical fiber splicer (Siemon, US-128 Ultrasplice) and the output fiber is connected to the photodetector or APD using a bare fiber terminator. After imaging, we disconnect fibers and return the mouse back to the breeding cage. The mouse can move freely without breaking the fibers. Supplementary Movie 2 shows a free-moving mouse wearing a usCCW.

**Research animals**. All animal procedures follow the guidelines established by the Use of Animals from the National Institutes of Health, as well as those set by the Institutional Animal Care and Use Committee of Northwestern University. All experimental protocols in this study were approved by the Animal Care and Use Committee of Northwestern University.

**Reporting summary**. Further information on research design is available in the Nature Research Reporting Summary linked to this article.

## Data availability

The data that support the findings of this study are available from the authors on reasonable request, see author contributions for specific data sets. All the SEM, optical, and Photoacoustic microscopy images in the figures and supplementary figures have been deposited in Open Science Framework online data repository (https://doi.org/10.17605/OSF.IO/DT8VE). The source data underlying Figs. 3e, h, i, 5d are provided as the Source Data file.

## Code availability

The custom code being used for image processing and analysis of this study are available from the authors on reasonable request, see author contributions for specific code.

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

## Acknowledgements

We thank Dr. Leonidas Ocola for his assistance in electron-beam nanolithography, and Dr. Ayush Batra and Dr. Neil Avadhoot Nadkarni for insightful advice and discussion on the hemorrhage and neovascularization observed during the long-term in vivo studies. We also thank Drs. Rui Cao and Song Hu for their assistance in mouse imaging. This work is supported by National Science Foundation grants DBI-1353952 and EEC-1530734; National Institutes of Health grants R01EY026078, R01EY029121, R01HL141933, and F30EY026472; a Research Catalyst Award by Northwestern McCormick School of Engineering, and a Northwestern University Innovative Initiative Incubator (I3) Award. Use of the Center for Nanoscale Materials, an Office of Science user facility, is supported by the U.S. Department of Energy, Office of Science, Office of Basic Energy Sciences, under Contract No. DE-AC02–06CH11357.

## Author contributions

H.L., B.D., H.F.Z. and C.S. initiated and designed the study. H.L., X.C. and D.C. fabricated the silicon master mold. H.L. and X.C. developed the sNIL method and fabricated MRR ultrasonic detectors. H.L., B.D. and R.H. packaged usCCWs. X.Z. and H.L. performed craniotomy for usCCW implantation. H.L., B.D. and X.S. conducted experimental characterizations and photoacoustic microscopy. B.D. and H.L. processed experimental data and imaging results. H.F.Z. and C.S. supervised the project. All authors discussed the results and contributed to the paper.

## Additional information

**Competing interests:** C.S. and H.F.Z. have financial interests in Opticent Health, which did not fund this work. The remaining authors declare no competing interests.

