## [Peer Review File · Nature Communications]

Reviewers' comments:

Reviewer #1 (Remarks to the Author):

This manuscript "Disposable ultrasound-sensing chronic cranial window by soft nanoimprinting lithography" introduces a new chronic cranial window (CCW) integrated with micro-ring resonator (MRR) that can detect ultrasound sensitively. Previously, the MRR-based ultrasound detector had disadvantages that the fabrication process was difficult and the life span was short in biological tissues. In this study, the authors overcome the problems by using a soft nano-imprint lithography method and a PDMS protection layer, and demonstrated longitudinal photoacoustic brain imaging through the CCW integrated with the MRR. The comments are below:

Major comments:

1. The major issue of this manuscript is that conventional PAM typically does not require any transcranial window for mouse cortical imaging, which is the biggest advantage over the conventional optical microscopic techniques. For longitudinal PAM imaging, no window is the best scenario and these results have been already published in many other journals over the last decade.
2. If the window is required for intervention, soft transcranial windows can be easily used and good for PAM imaging.
3. MRR technologies themselves are not really new because they have been reported for many years by the authors and other groups as well.

Technological comments

1. The sensitivity seems to be higher in the center than the periphery of the MRR. What is the effective field of view? It would be great if the author presents sensitivity depending on the distance from the center quantitatively.
2. The MRR detectors in References 12, 13, and 27 do not have a protective layer and their fabrication methods are different from the proposed detector in this manuscript. Are the bandwidth and noise equivalent pressures maintained the same in the new process detector? If not, their measured data would be required. In fact, the bandwidth in supplementary Fig. 3 seems to be shorter than 250 Mhz.
3. The Supplementary Fig. 5 is about the monitoring result not about the comparison of the sensitivity and frequency response with and without the PDMS protection layer. Please check the supplementary document. In addition, the Supplementary Fig. 6 is missing.
4. Is the PDMS soft mold also reusable?
5. On page 8, the authors said "To demonstrate that active CCW allows additional measurement capability other than optical access, we show that usCCW enabled longitudinal PAM cortical imaging (Figure 3a) for nearly one month."
Is CCW primarily not used for monitoring? In the first sentence of the introduction, the author clarified that CCW is commonly used for monitoring. Does that CCW mean passive CCW? If so, what does active CCW mean?
6. Functional imaging is defined as a medical imaging reflecting the changes in metabolism, blood flow, regional chemical composition, and absorption. However, the authors showed only the structural changes of vessels during its healing process. Therefore, it would be better to reconsider using this expression.

Reviewer #2 (Remarks to the Author):

I have reviewed the manuscript based on MRR device. The manuscript reported a novel method to fabricated the CCW and the performance of the device has been much improved compared with

the authors previous reports. My opinions are as below.

1, The authors said the 5-um thick PDMS cladding layer has negligible attenuation less than 0.6dB of ultrasound signal at 100MHz. The ultrasonic sensing work frequency of the PAM is 100MHz or other frequency?

2, What is the sensitivity of this MRR ?

3, The working bandwidth of this MRR is 250MHz? the author just said their previous device is working at 250MHz. Moreover, the characteristics and performance of the MRR provided authors are little, such as sensitivity, NEP, etc.

4, In the PAM experimental setup, More details are needed here including the gain used in the amplifier, the authors provided very little information for this in the manuscript.

On behalf of authors, we appreciate reviewers' comments. We have carefully revised our manuscript accordingly. While reviewer #2's comments are quite positive in general, we would like to express our concern regarding the reviewer #1's comments. Please see our point-by-point response to reviewers' comments below:

Reviewer #1

Major comment 1: The major issue of this manuscript is that conventional PAM typically does not require any transcranial window for mouse cortical imaging, which is the biggest advantage over the conventional optical microscopic techniques. For longitudinal PAM imaging, no window is the best scenario and these results have been already published in many other journals over the last decade.

Response: Unfortunately, this comment rather contradicts the governing physics principles in photoacoustic imaging. The intact mouse skull has long been a major technical hurdle for high-resolution photoacoustic brain imaging. The underlying scattering of the incident laser beam and attenuation of acoustic signal due to acoustic impedance mismatch respectively hampers both lateral and axial resolution of photoacoustic imaging. Specifically, the strong scattering of the incident laser beam unfavorably broadens the focal spot size and thus reduce the lateral resolution. On the other hand, the large acoustic impedance mismatch at the skull-brain interface leads to strong attenuation of the generated photo-acoustic signals, especially the high-frequency signals above 30 MHz (H. Estrada et al., *Physics in Medicine & Biology*, vol. 61, p. 1932, 2016, M. Kneipp et al., *J. Biophoton.*, vol. 9, p 117-123, 2016). As a result, the best reported PAM imaging results has a highest axial resolution of 30 μm using a 50 MHz ultrasound transducer. Such a low axial resolution poses fundamental challenges in differentiating capillary networks in the skull, the dural, and the cortex. In a recent study by J. Yao et. al., it has found that "The skull degraded the image quality of PAM by blurring the optical focusing and attenuating the PA signal" (*J. Yao et. al., Nature Methods, volume 12, pages 407–410, 2015*). In Yao's study, the intact mouse skull severely attenuated the acoustic signals, resulting in a ~70% signal loss at 50 MHz acoustic frequency. Such a strong acoustic attenuation makes it almost impossible to further improve the axial resolution by increasing the ultrasound frequencies above 50 MHz. Therefore, it is a consensus in photoacoustic imaging community that, the mouse skull is a major technical challenge that has prevented significant improvement in photoacoustic brain imaging.

To quantitatively illustrate the influence of the skull on the PAM image quality in adult mice, we also include here the recently performed comparative studies of PAM brain image with and without the skull (**Figure R1**). A ns laser ($\lambda = 532 \text{ nm}$) at the pulse repetition rate of 30 kHz was used as the source of excitation. For the convenience, a 30 MHz ultrasound transducer was used to detect the generated photoacoustic signals. As shown in **Figure R1a**, the PAM brain image was first performed in the adult mouse with intact skull, after surgically removing the scalp. The corresponding B-scan of dash line in Figure R1A is shown in **Figure R1c**. The strong optical scattering of incident ns-laser and the attenuation of the high-frequency ultrasound signals due to the presence of the skull unfavorably spoiled the recorded PAM brain images. For comparison, the skull of the same mouse was surgically removed, and the exposed region was subsequently covered with a transparent cranial window. **Figure R1b** shows the acquired PAM image through the cranial window, with the imaging resolution significantly improved in comparison with the PAM image through the intact skull shown in **Figure R1a**. Clearly, the use of cranial windows successfully eliminates scattering of the incident focused laser beam and thus, significantly improve the lateral resolutions of the acquired en-face PAM image (**Figure R1b**). The improved optical focusing also results in stronger ultrasound signals through a cranial window (**Figure R1d**), which is in clear contrast to the recorded ultrasound signals through the

intact skull (**Figure R1c**). Despite the greatly improved lateral resolution by eliminating the optical scattering, the axial resolution of PAM remains fundamentally limited to the ultrasound detection bandwidth. As shown in the x-z projection of the reconstructed images, the axial resolution with the cranial window (**Figure R1f**) was only marginally improved in comparison with the intact skull (**Figure R1e**).

Figure R1. The influence of the skull on the image quality is non-negligible in adult mice. A side-by-side comparison is shown below, where the PAM image acquired through the intact skull (**a**) shows much blurred microvasculature, in comparison to the acquired PAM image through a cranial window (**b**), due to the strong light scattering of the skull. (**c**) and (**d**) are the recorded B-scan ultrasound signals along the dashed lines marked in (**a**) and (**b**), respectively. The signals through a cranial window (**d**) are much stronger than through the intact skull (**c**). (**e**) and (**f**) are the x-z projection of the reconstructed B-scan PAM images shown in (**a**) and (**b**), respectively.

Thus, the focus of this study is to develop an active ultrasound sensing chronic cranial window (usCCW) that simultaneously overcomes all of the above-mentioned challenges caused by mouse skull by using optically transparent usCCW to improve the lateral resolution by eliminating the optical scattering, and integrating the microring resonator (MRR) ultrasound detector to the inner surface of usCCW to improve the axial resolution. The later was accomplish by (1) placing the MRR to the inner surface of usCCW to eliminate the attenuation due to the acoustic impedance mismatch, (2) significantly reducing the path length of ultrasound propagation to mitigate the strong attenuation of high-frequency ultrasound signal in tissue, and (3) the use of MRR ultrasound detector with significantly improved ultrasound detection sensitivity and bandwidth.

As the results, we demonstrated a smart cranial window based photoacoustic microscopy with an axial resolution of $\sim 4 \mu\text{m}$, which is the highest axial resolution ever achieved in photoacoustic brain imaging *in vivo*. Because our window itself is a wide-band ultrasound detector located at the close proximity to the brain tissue, the high axial resolution is, for the first time, achieved without sacrificing the optical focusing, the photoacoustic signal strength, or the field of view. The demonstrated 8-fold improvement in axial resolution makes it possible, for the first time, to clearly distinguish capillaries in the dura and cortex and thus, offers new opportunities for fundamental neuroscience investigations.

In addition, compared with our usCCW method, the reported PAM results through intact skulls also required surgically removing the scalp, making the studies similarly invasive. Moreover, in our study, we also evaluated the PAM brain imaging through thinned-skull. We found the regrow of the skull results in the degradation of the acquired PAM imaging quality, which would prevent longitudinal studies on the same animals with consistent imaging qualities (Supplementary information).

We believe that our work represents, so far, the best engineering solution to achieve longitudinal photoacoustic brain imaging with consistently high image quality. This work will provide the foundation for PAM to be much better integrated with modern brain studies.

Major comment 2: If the window is required for intervention, soft transcranial windows can be easily used and good for PAM imaging.

Response: The use of soft transcranial windows only partially solves the above-mentioned issues in photoacoustic brain imaging through intact skull. While it can effectively reduce the optical scattering as well as acoustic attenuation due to impedance mismatch, a bulky water tank is still required to couple the ultrasound signal to the externally-mounted ultrasound detector. The strong attenuation of soft transcranial window in high-frequency ultrasound signal (above 50 MHz) will significantly reduce the axial resolution for PAM brain imaging. Moreover, the high frequency attenuation by the coupling water layer is not negligible. So far, to the best of our knowledge, there has been no reported study implementing soft transcranial windows for long-term brain PAM imaging through water coupling. In contrast, we report here the first long-term photoacoustic brain imaging in live mouse, in an effectively non-contact manner. By concealing the ultrasound detection capabilities within the sealed usCCW, we eliminated the hurdle for long term photoacoustic microscopy for the need of bulky water tank for ultrasound coupling. Thus, we are confident that the report functional integration of MRR detector and the transparent cranial windows represents the best technical solution.

Major comment 3: MRR technologies themselves are not really new because they have been reported for many years by the authors and other groups as well.

Response: The usCCW reported here differs significantly from the MRR technologies being developed previously. As shown in Figure 3, the MRR device reported previously failed to work in 2 hours upon being exposed to blood. To ensure the reliable operation of usCCW suitable for long-term investigations, in this manuscript, we have implemented two major changes in the usCCW design and fabrication steps:

1. The micro-ring resonator is now encapsulated by an acoustic impedance-matched PDMS layer, which significantly extend the lifetime of the usCCW over the 28-day observation period (Fig. 3d).

2. Further, we developed soft nanoimprinting process for fabricating usCCW with significant cost saving and more importantly, more than 10-fold improved Q-factor. UsCCW maintains high Q-factor $> 3 \times 10^4$ over a 28-day observation period.

Technological comment 1: The sensitivity seems to be higher in the center than the periphery of the MRR. What is the effective field of view? It would be great if the author presents sensitivity depending on the distance from the center quantitatively.

Response: The miniaturized footprint of the MRR (80 μm in diameter) makes it an idea point ultrasound detector. It prompts greatly improved angular detection range in comparing with the conventional piezoelectric area detector. The photoacoustic detection sensitivity of MRR is in general a function of the distance between the MRR and ultrasound source and the detection angle. It has been investigated extensively in our previous publication [Z. Zhang, *et al.* "Theoretical and experimental studies of distance dependent response of micro-ring resonator-based ultrasonic detectors for photoacoustic microscopy", *J. Appl. Phys.*, **116**, 144501 (2014)]. As also investigated in this reference, the field of view is determined by the distance between MRR and ultrasound source. In this work, the distance between the MRR and cortex is about 1 mm, corresponding to an effective field of view of 360 μm at 166 MHz bandwidth. Extended field of view of 600 μm can be obtained at 100 MHz bandwidth.

We added the field of view on page 9, paragraph 2.

Technological comment 2. The MRR detectors in References 12, 13, and 27 do not have a protective layer and their fabrication methods are different from the proposed detector in this manuscript. Are the bandwidth and noise equivalent pressures maintained the same in the new process detector? If not, their measured data would be required. In fact, the bandwidth in supplementary Fig. 3 seems to be shorter than 250 Mhz.

Response: In this study, we have developed a completely new fabrication process flow to ensure reliable operation of MRR detector of long-term in vivo studies, which fundamentally differs from the device we reported previously in References 12, 13, and 27. As the results, we have achieved 14-fold improvement in Q-factor (1.46×10^5), which positively contributes to the increasing in the detection bandwidth to 250 MHz as well as reduction in the NEP to 0.49 Pa.

For reference, we have included the detailed explanation as follows:

MRR detectors shown in Reference 12, 13, and 27 were fabricated using direct ebeam lithography process without the protection layer, with the highest Q-factor of 10,400 being recorded experimentally. The sensitivity of MRR can be defined as:

$$Sensitivity = \frac{dT}{dP} = \frac{dn_{eff}}{dP} \left(\frac{d\phi}{dn_{eff}} \frac{dT}{d\phi} \right)_{\phi_0},$$

where P is the ultrasonic pressure; T is the transmission through the bus waveguide; n_{eff} is the effective refractive index; and ϕ_0 is the phase bias at the resonance wavelength. These three terms dn_{eff}/dP , $d\phi/dn_{eff}$, and $dT/d\phi$ are collectively determined by the quality factor (Q-factor) of the MRR. Hence improving the Q-factor will favorably benefit ultrasound detection sensitivity. Considering the experimentally measured noise floor of 0.32 mV, the corresponding noise equivalent pressures (NEP) is 6.8 Pa.

To ensure the proper operation of MRR detector in the demanding in vivo environment, we developed a completely new soft nano-imprinting lithography (**sNIL**) process for fabricating the MRR detectors with the emphasis on improving the sensitivity and reliability (Figure 1 in the manuscript). The Q-factor of MRR is inversely proportional to the propagation loss and

scattering loss inside the micro-ring waveguide, and its coupling loss with the bus waveguide. The overall loss reduction was accomplished in three steps: (1) Propagation loss reduction: We chose polystyrene as the waveguide core material with 10-fold loss reduction in comparison with the previously used SU-8 materials [17]; (2) Scattering loss reduction: thermal oxidation and wet etching processes are implemented to reduce surface roughness of the silicon master mold and thus the scattering loss; (3) Coupling loss reduction: DRIE etching process allows fabrication of a high-aspect-ratio gap between the bus and micro-ring waveguide with high fidelity. It enables us to reduce the coupling loss in alignment with the much-reduced intrinsic loss, to reach the critical coupling condition for maximizing the Q-factor of MRR.

As the results, the newly established sNIL promises greatly improved Q-factor of 1.46×10^5 (**Figure R2a**), which corresponds to more than 14-fold improvement than previous results. The ultrasonic sensitivity of MRR is expected to be 391.1 V/MPa. Considering the noise floor of 0.32 mV in our previous studies, the NEP is estimated to be 0.49 Pa. Additionally, the proposed sNIL process significantly reduces the unit cost of MMR to be less than a dollar. Finally, encapsulating MRR device by a 5- μm -thick PDMS thin protection layer help to maintain the reliable operation of MRR for long-term investigation. **Figure R2b, c, e, & f** show MRR with and without these protections when exposed to undiluted blood. Without protection, the resonance diminished in 2 hours (**Figure R2d**), while the Q-factor of the protected ones remained unperturbed (**Figure R2g**).

In the revised manuscript, we added the newly estimated noise equivalent pressure on page 7, paragraph 1 and the estimation details in Supplementary Note 4. We added more experimental details in supplementary Figure S4.

Figure R2. (a) Transmission spectrum of the MRR with a FWHM of 0.0096 nm, showing a Q-factor of 8.6×10^4 . (b) Schematic and (c) SEM images of the unprotected MRR. (d) Optical resonance diminishes in two hours. (e) Schematic and (f) SEM image of the MRR with protection layer. (g) Q-factor remained unaffected after being exposed to whole blood for two hours. Scale bars in (c),(f): 500 nm.

The bandwidth shown in Supplementary Figure 3 (now Supplementary Figure S4) is indeed narrower than 250 MHz, which is mainly due to the limited bandwidth of the low-noise APD photodetector (Hamamatsu C4777) being used. The calculated 6-dB bandwidths in Supplementary Figure S4 are 166.4 MHz for MRR without protection layer and 164.9 MHz for the one with PDMS protection layer.

We have added additional explanation in the revised manuscript as well as supplementary materials.

Technological comments 3: The Supplementary Fig. 5 is about the monitoring result not about the comparison of the sensitivity and frequency response with and without the PDMS protection layer. Please check the supplementary document. In addition, the Supplementary Fig. 6 is missing.

Response: We apologize for the errors in the manuscript. They have been corrected in the revised manuscript.

Technological comments 4: Is the PDMS soft mold also reusable?

Response: Yes, the PDMS soft mold can be reused for 3 to 5 times. We added the corresponding descriptions on page 4, paragraph 2 in the manuscript and page 18, paragraph 3 in the method.

Technological comments 5: On page 8, the authors said “To demonstrate that active CCW allows additional measurement capability other than optical access, we show that usCCW enabled longitudinal PAM cortical imaging (Figure 3a) for nearly one month.”

Is CCW primarily not used for monitoring? In the first sentence of the introduction, the author clarified that CCW is commonly used for monitoring. Does that CCW mean passive CCW? If so, what does active CCW mean?

Response: The “active CCW” mentioned here refers to the usCCW that integrates the MRR ultrasound detector as the active sensing elements. We have therefore revised the noted sentence as follows to avoid confusion:

“By integrating the MRR ultrasound detector as the active sensing elements, we show that usCCW enabled longitudinal PAM cortical imaging (Figure 3a) for nearly one month.”

Technological comments 6. Functional imaging is defined as a medical imaging reflecting the changes in metabolism, blood flow, regional chemical composition, and absorption. However, the authors showed only the structural changes of vessels during its healing process. Therefore, it would be better to reconsider using this expression.

Response: We do agree with the reviewer that we only showed the morphological changes in the cortical vasculatures. But after carefully checking the manuscript, we don't believe we have ever made the claim of functional imaging in this manuscript.

Reviewer #2 (Remarks to the Author):

I have reviewed the manuscript based on MRR device. The manuscript reported a novel method to fabricated the CCW and the performance of the device has been much improved compared with the authors previous reports. My opinions are as below.

Comment 1: The authors said the 5-um thick PDMS cladding layer has negligible attenuation less than 0.6dB of ultrasound signal at 100MHz. The ultrasonic sensing work frequency of the PAM is 100MHz or other frequency?

Response: As shown in Supplementary Figure S4, fabricated CCW is operated at the 6-dB bandwidth from 10 kHz to 164.9 MHz, which is mainly due to the limited bandwidth of the low-noise APD photodetector (Hamamatsu C4777) being used.

We have included the detailed description of the detection bandwidth on page 20, paragraph 1 (method section).

Comment 2: What is the sensitivity of this MRR?

Response: The sensitivity of MRR can be defined as:

$$Sensitivity = \frac{dT}{dP} = \frac{dn_{eff}}{dP} \left(\frac{d\phi}{dn_{eff}} \frac{dT}{d\phi} \right)_{\phi_0},$$

where P is the ultrasonic pressure; T is the transmission through the bus waveguide; n_{eff} is the effective refractive index; and ϕ_0 is the phase bias at the resonance wavelength. These three terms dn_{eff}/dP , $d\phi/dn_{eff}$, and $dT/d\phi$ are collectively determined by the quality factor (Q-factor) of the MRR. With the experimentally measured Q-factor of 1.46×10^5 , the ultrasonic detection sensitivity of MRR is estimated to be 391.1 V/MPa. Considering the noise floor of 0.32 mV in our previous studies, the NEP is estimated to be 0.49 Pa.

We have included the discussion of sensitivity and NEP on page 7, paragraph 1.

Comment 3, The working bandwidth of this MRR is 250MHz? the author just said their previous device is working at 250MHz. Moreover, the characteristics and performance of the MRR provided authors are little, such as sensitivity, NEP, etc.

Response:

As shown in Supplementary Figure S4, fabricated CCW is operated at the 6-dB bandwidth from 10 kHz to 164.9 MHz, which is mainly due to the limited bandwidth of the low-noise APD photodetector (Hamamatsu C4777) being used. As suggested by the reviewer, we have included additional discussion on sensitivity, NEP, bandwidth, and field of view in the revised manuscript. The detailed explanation has also been added in the supplementary information as supplementary note 4.

Comment 4, In the PAM experimental setup, more details are needed here including the gain used in the amplifier, the authors provided very little information for this in the manuscript.

Response: We have carefully revised the manuscript with additional details of the experimental setup. It has been included in the supplementary note 4 in the revised supplementary information.

REVIEWERS' COMMENTS:

Reviewer #1 (Remarks to the Author):

We think that the authors well addressed our previous technological comments. However, we still doubt whether the novelty and significance of this paper is enough to be published in this journal.

As for the authors' responses to previous major comments #1–#2, we agree that the usCCW can improve image quality. However, the effect comes from the cranial window, which is already proven. Conventional PAM can detect the major vessels through intact skulls, as shown in Figure R1a, and can acquire the improved images with the conventional cranial windows, as shown in Figure R1b. Therefore, we doubt whether the proposed usCCW can have any significant impact or improvement compared to the conventional PAM although there is a little improvement.

Reviewer #2 (Remarks to the Author):

The authors have revised the manuscript based on the review comments. The authors has clearly demonstrate their novelty and improvement compared with previous reported PAM. I have no comments at this stage.